# Web-MCOT Server for Motif Co-Occurrence Search in ChIP-Seq Data

**DOI:** 10.3390/ijms23168981

**Published:** 2022-08-11

**Authors:** Victor G. Levitsky, Alexey M. Mukhin, Dmitry Yu. Oshchepkov, Elena V. Zemlyanskaya, Sergey A. Lashin

**Affiliations:** 1Department of System Biology, Institute of Cytology and Genetics, 630090 Novosibirsk, Russia; 2Department of Natural Science, Novosibirsk State University, 630090 Novosibirsk, Russia

**Keywords:** chromatin immunoprecipitation with massively parallel sequencing, transcription factors binding sites prediction, co-binding of transcription factors, composite elements, transcription factor binding sites, motifs conservation, overlap of motifs

## Abstract

(1) Background: The widespread application of ChIP-seq technology requires annotation of cis-regulatory modules through the search of co-occurred motifs. (2) Methods: We present the web server Motifs Co-Occurrence Tool (Web-MCOT) that for a single ChIP-seq dataset detects the composite elements (CEs) or overrepresented homo- and heterotypic pairs of motifs with spacers and overlaps, with any mutual orientations, uncovering various similarities to recognition models within pairs of motifs. The first (Anchor) motif in CEs respects the target transcription factor of the ChIP-seq experiment, while the second one (Partner) can be defined either by a user or a public library of Partner motifs being processed. (3) Results: Web-MCOT computes the significances of CEs without reference to motif conservation and those with more conserved Partner and Anchor motifs. Graphic results show histograms of CE abundance depending on orientations of motifs, overlap and spacer lengths; logos of the most common CE structural types with an overlap of motifs, and heatmaps depicting the abundance of CEs with one motif possessing higher conservation than another. (4) Conclusions: Novel capacities of Web-MCOT allow retrieving from a single ChIP-seq dataset with maximal information on the co-occurrence of motifs and potentiates planning of next ChIP-seq experiments.

## 1. Introduction

Commonly, two or more transcription factors (TFs) work coordinately to induce transcriptional change [1]. At the genome level, this leads to composite elements (CEs) as specific overrepresented combinations of two motifs in regulatory regions of genes. Two motifs may overlap each other, or a spacer separates them [2]. Chromatin immunoprecipitation followed by massive sequencing (ChIP-seq) specifically for given cells, tissue type and condition maps peaks in a genome and motivates the genome-wide search of binding sites (motifs) not only for the target (Anchor) TF but also for co-binding (Partner) TFs. Several recently developed databases provide uniformly processed genomic profiles for thousands of ChIP-seq experiments [3,4,5]. Widespread direct interactions of many TFs with DNA in chromatin cause the significant overrepresentation of their motifs in deduced peaks. Conventionally, a *de novo* motif search [6,7] may define among them the motif of Anchor TF, but the enrichment of other motifs only point to many possible Partner TFs. Thus, the CE search task for ChIP-seq data consists in the detection of overrepresented motifs of Partner TFs located nearby the Anchor motifs. The web tool SpaMo represented the first approach to detect the motif co-occurrence in a single ChIP-seq dataset [8]. It searched CEs only with spacers, though later studies indicate that CEs with overlaps of motifs are much more common than CEs with spacers [9,10]. Another approach [11,12] revealed CEs with spacers and overlaps of motifs through the analysis of a special benchmark collection of multiple consistent ChIP-seq datasets. Unfortunately, due to the impossibility of many datasets being included in such consistent collections, this approach has not been widely applied and it was not implemented as a web tool. Accordingly, up to now, there are no web tools for predicting the significantly enriched co-occurrence of overlapped and spaced motifs in a single ChIP-seq dataset (Table 1).

We recently developed the Motifs Co-Occurrence Tool (MCOT), which integrated the prediction of CEs with spacers and overlaps for a single ChIP-seq dataset [10,13]. In this study, we propose the web tool Web-MCOT promoting this approach. For a given Anchor motif and a dataset of input peaks, Web-MCOT tests the significance of CE overrepresentation for a user-defined Partner motif, or it checks the significance of CEs with a variety of Partner motifs from a library derived from genome-wide sequencing experiments [14,15]. Web-MOCT results allows the sorting of output data; hence, the structure and abundance of top-scoring CEs explain the impacts of Partner TFs supporting the interaction of an Anchor TF with DNA in chromatin.

## 2. Results

### 2.1. Input Data

The homepage of Web-MCOT represents the common description of the web server and proposes to proceed to data analysis (Figure 1A, the link ‘Application’). The trigram symbol ‘≡’ in the top left corner (Figure 1A) hides the menu and extends the input or output data page up to the whole width of screen. The application page (Figure 1B) requires the following input data: sequences of peaks (the option ‘Upload or Enter DNA sequences in FASTA format’); a nucleotide frequency matrix of an Anchor motif (the option ‘Upload or Enter Anchor motif’); a single Partner motif or a library of Partner motifs (the option ‘One or Many Partner motif(s) will be tested’).

A *de novo* motif search in peak sequences is the best option since it directly reflects the nucleotide context inherent to the Anchor motifs. Thus, the motifs enriched in ChIP-seq peaks are stored in a number of databases [3,4,5]. One of them, the Cistrome DB [3], provides ChIP-seq peaks integrated with the results of *de novo* motif search, i.e., enriched motifs as nucleotide frequency matrices. Alternatively, matrices respecting Anchor motifs can be derived indirectly from Hocomoco [14], CIS-BP [16], or JASPAR [17] databases. The ‘Many Partners’ option implies the setting of a public database, i.e., for one Anchor motif, many Partner motifs are tested at once. The Hocomoco database for mammals [14] represents the human/mouse core and full collections of 396/353 and 747/509 motifs, or the Plant Cistrome collection provides 514 motifs for *A. thaliana* [15]. The advanced option (Figure 1B) allows changing the limits of a spacer length. Pressing the ’RUN’ button starts the calculation and provides a web link to the results. During a running process, the special indicator shows the percentage of completed calculations. The computational time may vary from several minutes to a couple of hours depending on the number of peaks and their average length. The example input data illustrate the Web-MCOT functionality (Figure 1B, button ‘EXAMPLE’). The example page (Figure 1C) contains peaks and the Anchor motif of the ChIP-seq dataset for FoxA2 from mouse liver [18].

### 2.2. Output Data

Figure 2 shows the Web-MCOT basic output table for the example FoxA2 dataset. The table rows respect second motifs in CEs, since all CEs have the same first Anchor motif. Hence, the first column ‘Motif name’ lists the names of the second motif. The name ‘Anchor’ would mean homotypic CEs, otherwise the names of Partner motifs respect heterotypic CEs. Next, the common header ‘CE significance, −Log10(*P-value*)’ joins five columns showing significances of CEs without respect to motif conservation respecting five computation flows. The next column, ‘Similarity, −Log10(*P-value*)’, marks the significances of similarity between Anchor and Partner motifs. The column ‘CE histogram’ displays icons for distributions of the abundance of CEs as a function of mutual orientation and location of motifs. These icons are links to larger-sized histograms. Figure 3 shows this histogram for the example CE FoxA2/HNF1B. For each Partner motif, among all CEs with overlaps of motifs, we find the most common one and draw a CE logo; see icons in the column ‘CE logo’ (Figure 2). They are links for CE logos. Figure 4A shows the logo for the CE example FoxA2/HNF1B. This logo explains how two motifs with a specific orientation and overlap are located in almost the same place. Finally, we provide links to heatmaps that show the abundance of CEs with various ratios of conservation between Anchor and Partner motifs. Five columns with the common header ‘Asymmetry heatmap, per mille’ contain links to these heatmaps for five computation flows. The example heatmap in Figure 4B shows that in overlapped locations of FoxA2 and HNF1B motifs, the second one possesses higher conservation than the first. The output table allows sorting the rows in ascending or descending order, all columns containing text or numerical content. We marked this option with the up/down tooltip arrows appearing next to the column headings. Thus, the example (Figure 2) represents the sorting for the column ‘Asymmetry, −Log10(*P-value*)’, ‘Overlap’.

Four links above the output table (Figure 2) allow download input and all output data. Thus, the link ‘Download *P-value* table’ (Appendix A) lists significances for the example CEs FoxA2/HNF1B; the link ‘Download histogram data’ (Appendix A) contains the source data for histograms of the abundance for CEs between the Anchor FoxA2 and all Partner motifs, so that for each histogram, various orientations, overlaps and spacers are compared. The link ‘Download additional data’ contains the rest output data, e.g., the list predicted CEs FoxA2/HNF1B for all peaks of the example FoxA2 dataset (Appendix A). The manual page (the link Help, Figure 1A) describes all output data in detail.

### 2.3. Architecture

The web server contains the frontend, work processes, and backend parts. The frontend part builds Single Page Application using JavaScript language, HTML, CSS and Vue.JS framework. The work-processes part performs kernel and additional calculations. The C++ kernel maps CEs in peaks and computes their significances. Kernel source code is available at the stand-alone MCOT site [19]. Additionally, R script generates heatmaps of CEs abundance with various conservation of motifs; Python code draws histograms of structural variants of predicted CEs with various orientations, overlaps and spacers; and we used the standard WebLogo library [20] to develop the library [21] producing the logo for CEs. The backend part is developed in Python language with Flask framework to build REST API web-service, Celely and Redis databases organize the task query and uWSGI application server.

## 3. Discussion

Our previous study [10] confirmed that MCOT is a methodologically novel approach for CE prediction, which outperforms in the recognition performance other available tools (see Table 1). Here we constructed the Web-MCOT server detecting significantly enriched co-occurred motifs taking into account their overlapping and spacing in a single ChIP-seq dataset. The server extends advantages of previously developed tools [8,11,12] for motif co-occurrence prediction in ChIP-seq data. Web-MCOT is similar to the popular web tool SpaMo [8], since for given Anchor motif it requires only a single ChIP-seq dataset, tests the co-occurrence of spaced motifs, and reveals Partner motifs with top-ranked significances of co-occurrence by processing a library of Partner motifs. The advantage of Web-MCOT consists in analysis of the co-occurrence of motifs with overlaps, and in analysis of co-occurrence of motifs with various conservations in pairs. The significant co-occurrence of motifs with an overlap may suggest synergistic or antagonistic mechanisms of their cooperation. The systematic difference in the conservation between two co-occurring motifs proposes that the TF with more conserved motif potentiates TF-DNA interaction of another TF of a pair [13]. Various visualization opportunities of Web-MCOT including histograms depicting abundance of CEs with various mutual orientations, overlaps and spacers of motifs, and the logos for the most common CE structural types with an overlap of motifs may further clarify the interaction between two respective TFs and genomic DNA.

## 4. Materials and Methods

### 4.1. Algorithm

Web-MCOT uses the previously described algorithm [10,13] to compute for a single ChIP-seq dataset and pairs of motifs the significances of co-occurrence for their locations with overlaps and with spacers. Web-MCOT takes the following input data: DNA sequences of ChIP-seq peaks, an Anchor motif, and a Partner motif or a name of Partner motifs library (Figure 5A). The homotypic and heterotypic pairs of motifs are considered, i.e., containing the same (Anchor/Anchor) or distinct (Anchor/Partner) motifs. Web-MCOT applies the recognition model of Position Weight Matrix for mapping motifs in sequences. For each heterotypic CE, Web-MCOT computes the significance of similarity between Anchor and Partner motifs, *P-value*(A,P) [10]. The criterion *P-value*(A,P) < 0.05 means that a CE may be false-positive prediction. For each motif, Web-MCOT computes five recognition profiles respecting five ranges of conservation levels (CL) deduced from estimates of the false-positive rate (FPR), CL = −Log10(FPR). FPRs are estimated as recognition rates for the whole-genome dataset of promoters. For each pair of motifs and each foreground sequence, Web-MCOT performs a permutation procedure to generate the background sequences. Next, the tool counts CEs in the foreground and background datasets and classifies CEs according to the location, orientation, similarity and conservation of participant motifs.

### 4.2. Mutual Orientations and Mutual Locations

Four types of mutual orientation of motifs include Direct AP (Direct PA) types respecting shifted location of Anchor/Partner (Partner/Anchor) motifs in the same DNA strand; the opposite DNA strand allows Everted and Inverted types (Figure 5B).

Five types of mutual location of motifs comprise ‘Full overlap’ denoting one motif entirely covering another, ‘Partial overlap’ meaning all other overlaps, ‘Overlap’ respecting any overlap, ‘Spacer’ representing spacing of motifs, and ‘Any’ designating either Overlap or Spacer (Figure 5C). Hence, we use five separate computation flows: Full, Partial, Overlap, Spacer, and Any. For all flows, we count sequences with/without CEs in foreground and background datasets and compute the significance *P-value* (CE) with Fisher’s exact test (Figure 5E).

### 4.3. Conservation of Motifs

Estimates of CL divide all CEs into two classes of those with more conserved Anchor or Partner motifs (Figure 5D). Besides two separate significances {*P-value* (CE)} for these classes, we compute the significance of asymmetry within CEs (Figure 5E,F). We assigned to this significance –Log10(*P-value*) signs ‘+’/‘−’ in the cases of the enrichment toward an Anchor/Partner motif. To draw the asymmetry heatmap for an Anchor/Partner pair, we consider CLs in the ranges CL < 3.5, 3.5 ≤ CL < 3.7, etc., up to 5.3 ≤ CL < 5.5 and CL ≥ 5.5, and we count CEs with specific CLs for foreground and background datasets. The heatmap shows the per-mille measure, {1000 × Obs_i,j_/Obs} − {1000 × Exp_i,j_/Exp}. Here, Obs_i,j_ and Exp_i,j_ mean counts of CEs for specific CLs for the foreground and background datasets, respectively; Obs and Exp denote corresponding total counts; indices i and j refer to Anchor and Partner motifs.

## 5. Conclusions

In this study, we propose the web server Web-MCOT, which predicts pairs of co-occurring spaced or overlapped motifs in a single ChIP-seq dataset. Web-MCOT requires DNA sequences of ChIP-seq peaks, a motif of the target TF (Anchor motif) and a motif of potential Partner TF (or a name of a public library of potential Partner motifs). Web-MCOT checks the significance of overrepresentation for pairs of overlapped and spaced motifs and visualizes the most significant pairs of motifs, taking into account various orientations and conservations of participants. Web-MCOT results may uncover various mechanisms of interaction between a target TF and genomic DNA, facilitating the planning of future ChIP-seq experiments.

## Figures and Tables

**Figure 1 ijms-23-08981-f001:**
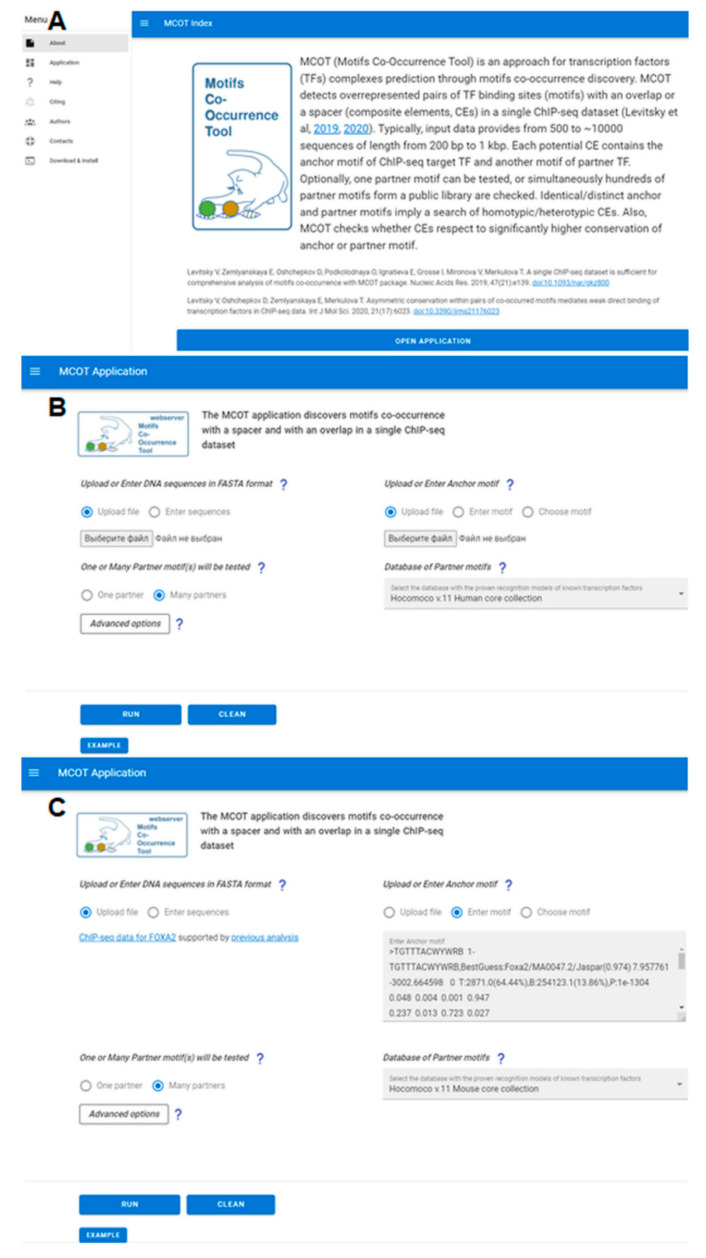
Basic Web-MCOT pages: (**A**) homepage, (**B**) application, and (**C**) example.

**Figure 2 ijms-23-08981-f002:**
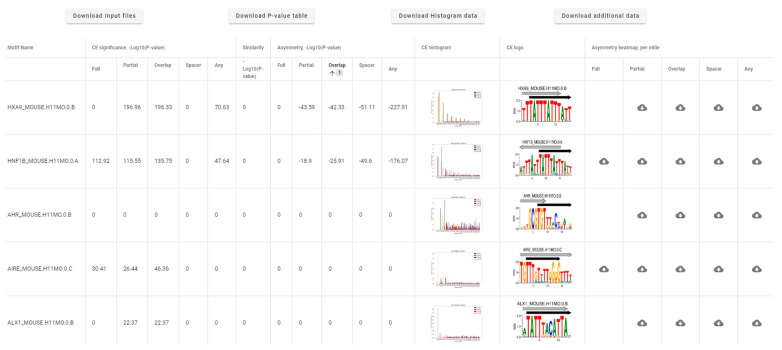
Basic output table of Web-MCOT for the example FoxA2 ChIP-seq dataset [18]. The first column lists the names of the second motifs in CEs. The common header ‘CE significance, −Log10(*P-value*)’ joins columns showing the significances of CEs without respect to motif conservation for five computation flows. The column ‘Similarity, −Log10(*P-value*)’ shows the significance of similarity of motifs. The common header ‘Asymmetry, −Log10(*P-value*)’ joins columns with the significances of asymmetry within CEs for five computation flows. The columns ‘CE histogram’ and ‘CE logo’ contain icons for histograms of CE abundance as a function of mutual orientation and location of the motifs, and for logos for the most common CE structural type with an overlap of motifs. The last five columns with the common header ‘Asymmetry heatmap, per mille’ respect five flows and show the links to heamaps of CEs abundance as a function of the ratios of conservation between Anchor and Partner motifs.

**Figure 3 ijms-23-08981-f003:**
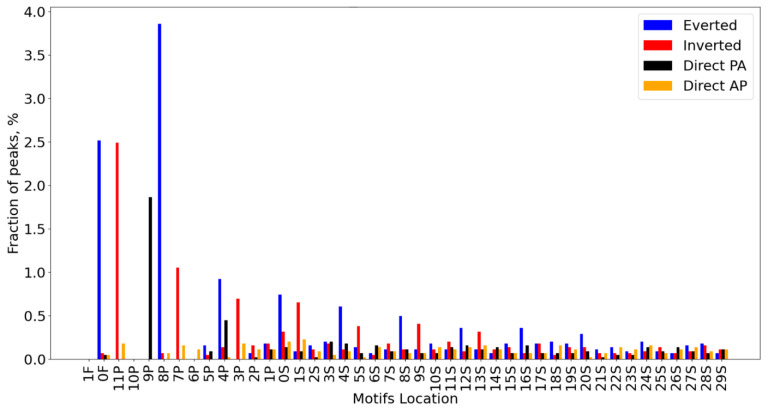
Distribution of structural variants of predicted CEs with various orientations, overlaps and spacers of the Anchor FoxA2 and Partner HNF1B motifs for the FoxA2 dataset [18]. Colors denote different mutual orientations. The letters in labels near axis X from left to right mean full (‘F’) or partial overlaps (‘P’), and spacer length (‘S’). The numbers preceding these letters denote the distance between nearest borders of two motifs, the length of overlap and the length of spacer, respectively. Axis Y shows the percentage of peaks containing CEs variants specific in mutual orientation and location. FoxA2 and HNF1B motifs were derived with the Homer tool [6] and the mouse Hocomoco library [14], respectively.

**Figure 4 ijms-23-08981-f004:**
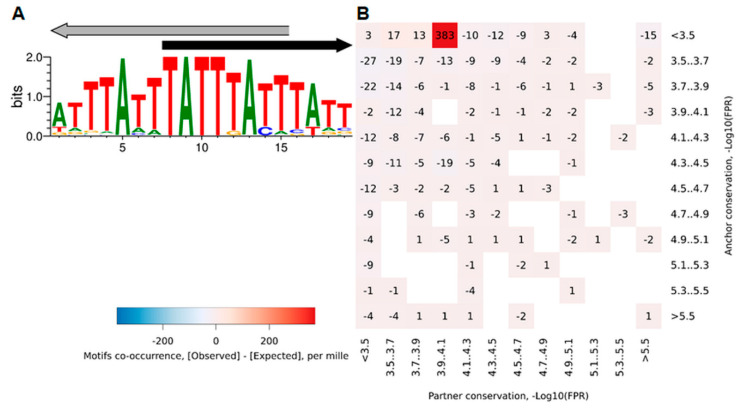
Graphical description of the most common CEs with an overlap of the Anchor FoxA2 and Partner HNF1B motifs for the FoxA2 dataset [18]. (**A**) Logo of the most common CE structural type. Black/grey arrows show the location and orientation of Anchor/Partner motifs. (**B**) Heatmap visualization of relationship of motifs conservation in CEs. Axes X/Y show ranges of conservation level of Partner/Anchor motifs. The color implies the per-mille measure for difference between observed and expected abundances of CEs with specific conservation of motifs (see Section 4.3).

**Figure 5 ijms-23-08981-f005:**
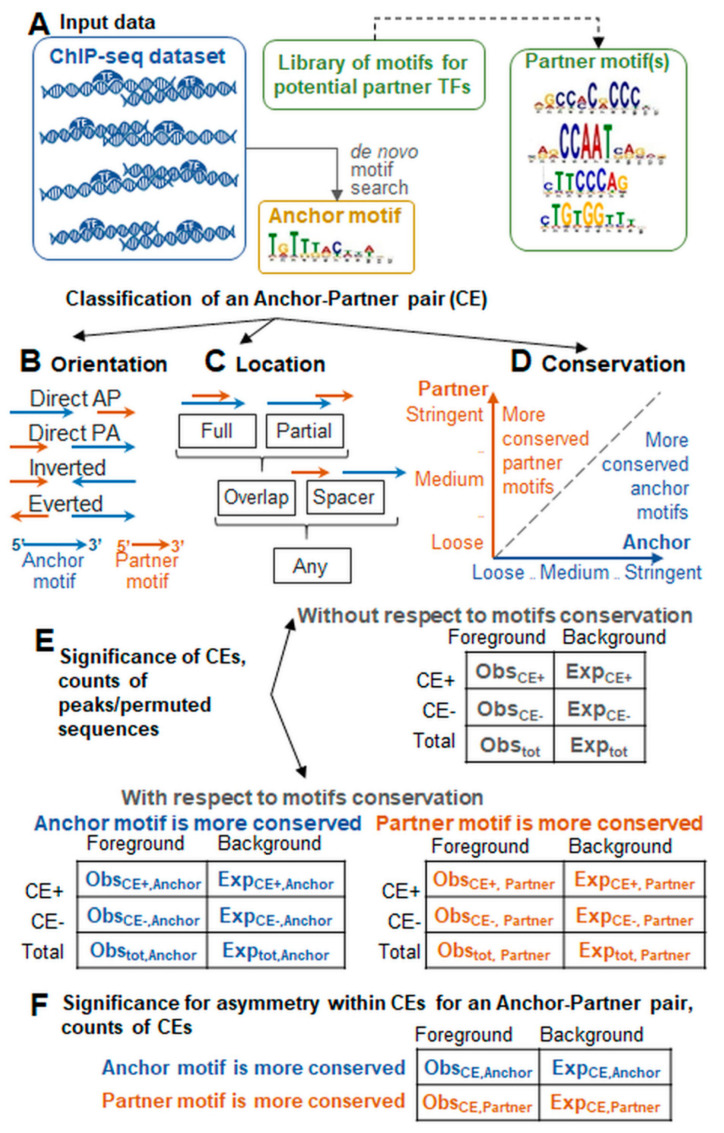
Web-MCOT workflow details. (**A**) Input data comprise a dataset of ChIP-seq peaks; a nucleotide frequency matrix for an Anchor motif; and a nucleotide frequency matrix for a Partner motif or designation of a public library of partner motifs. (**B**–**D**) show CEs classification according to orientations, locations (overlaps and spacers), and ratios of motifs conservation. (**E**) 2 × 2 contingency tables for computation of significances of enrichment for three CE types: without taking into account relationships of motifs conservation, and with an Anchor/Partner motif possessing higher conservation than Partner/Anchor motif (notations CE+/CE− respect sequences with/without CEs). (**F**) 2 × 2 contingency table for computation of significances of asymmetry within CEs.

**Table 1 ijms-23-08981-t001:** Comparison of various tools for prediction of CEs in ChIP-seq data.

Tool Name	A Single Dataset of Peaks Is Sufficient	Overlapped Motifs Are Allowed	URL	Reference
SpaMo	Yes	No	https://meme-suite.org/meme/tools/spamo (accessed on 9 August 2022)	[8]
TACO	No	Yes	http://bioputer.mimuw.edu.pl/taco/ (accessed on 9 August 2022)	[12]
MCOT, Web-MCOT	Yes	Yes	https://github.com/AcaDemIQ/mcot-kernel, https://webmcot.sysbio.cytogen.ru (accessed on 9 August 2022)	[10], this study

## Data Availability

Web-MCOT is available online at https://webmcot.sysbio.cytogen.ru/ (accessed on 18 July 2022).

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
