# Peer review of "Web-MCOT Server for Motif Co-Occurrence Search in ChIP-Seq Data"

_ijms, 2022, doi:10.3390/ijms23168981_

Round 1

Reviewer 1 Report

Dear Author 

The manuscript titled "Web-MCOT Server for Motifs Co-Occurrence Search in ChIP-seq Data" is significant to the field.

I would add literature searches from published data to make the MCOT attractive.

I would like to know the quality of data sets which could be a limiting factor.

Peak calling depends on parameters and software used, and MCOT seems ultimately dependent on peak calls; how to rectify this, or are there any suggestions?

Best

Author Response

Attached file shows the detailed point-by-point response to the reviewer’s comments

Reviewer 2 Report

In this work Levitsky et al. have presented the web server (Web-MCOT) for an already published tool for determining motif co-occurrence (MCOT) from a single ChIP-seq dataset. This web server is useful for researchers as it helps in determining anchor motifs and partner motifs for a chip-seq dataset in a user friendly manner. However there are several major concerns that needs to be addressed.

1. Authors must provide a table which should show the comparison of Web-MCOT with all existig motif co-occurrence tools. This table should focus on the features that outstand Web-MCOT with respect to the other tools.

2. Is it possible to check more than one anchor motif simultaneously in the web server? What is the maximum number of partner motifs that can be checked?

3. As the input fasta, do users need to provide the sequence of the entire peak or the sequence of the summit is enough to determine the partner motif? Is there a way that if summit sequences are provided the tool can extend the window and scan a specific region?

4. Is it possible to input just the anchor motif sequence instead of a nucleotide frequency matrix? It might be possible that the matrix is not available to an user. What can be the best possible way to use the tool in this scenario?

Author Response

(The authors gave the same response as above.)

Round 2

Reviewer 2 Report

Authors have satisfactorily responded to all comments.